# Surgical Trimming: Minimal Sufficient Chain of Thought with RazorReward-RL

## Abstract

Recent advances in chain-of-thought (CoT) and post-training have improved LLMs' reasoning abilities, but often at the cost of generating redundant steps, leading to wasted computation and increased latency in real-time applications. Existing reinforcement learning (RL) approaches attempt to condense CoT by rewarding brevity, but they fall short in two key aspects: (1) For highly difficult queries, they waste tokens on hopeless reasoning attempts; (2) For medium-difficulty queries, models either stop too soon and miss the answer, or continue beyond the correct answer and introduce errors. To address these issues, we propose RazorReward—a novel reward scheme that sharply differentiates optimal from suboptimal reasoning. For hard queries, RazorReward penalizes unnecessary CoT steps and encourages abstention when no solution is possible. For medium-difficulty queries, it rewards only reasoning paths that match the minimal sufficient CoT steps, heavily penalizing both under- and over-reasoning. Building on this, we introduce RazorReward-RL, a novel RL framework that segments CoT into semantically meaningful blocks, enabling more precise early stopping and targeted reward allocation. Extensive experiments on six reasoning benchmarks show that RazorReward-RL consistently outperforms previous methods, boosting accuracy by 8.3%–9.3% while reducing average token usage by 38.4%–43.8%, thus achieving a better balance between accuracy and efficiency.

## 1 Introduction

Driven by advances in chain-of-thought (CoT) prompting (Wei et al., 2022; Dai et al., 2025a), Large Language Models (LLMs) have demonstrated remarkable progress on complex reasoning tasks (Liu et al., 2025; Li et al., 2025; Chen et al., 2024b). Notable models like DeepSeek-R1 (DeepSeek-AI et al., 2025) and QwQ (Team, 2025) exhibit exceptional performance across diverse benchmarks. Despite these advancements, such reasoning models often suffer from a critical limitation known as *overthinking*. As first identified by (Chen et al., 2024a), *overthinking* occurs when LLMs generate unnecessarily verbose or redundant reasoning steps, even for easy queries. This leads to excessive token usage and slower responses, which can harm user experience in latency-sensitive applications like search engines (Team et al., 2025).

To minimize redundant reasoning, recent studies leverage Reinforcement Learning (RL)(Shao et al., 2024; Bai et al., 2022; Ouyang et al., 2022; Ramesh et al., 2024) to align LLM outputs with condensed CoT reasoning (Team et al., 2025; Hou et al., 2025; Yi & Wang, 2025). The key difference among these RL-based methods lies in how rewards are constructed for positive and negative samples. To clarify this, we adopt the sampling framework utilized in S-GRPO (Dai et al., 2025b): they truncate the generated CoT sequence at predefined positions (e.g., token index $p_i$) while appending an early-stopping prompt (e.g. "*Thinking time is up; please output the required answer*"). The model is then prompted to generate an answer from this shortened CoT, with correct answers labeled as positive samples and incorrect ones as negative samples. Based on this framework, we categorize input queries into three difficulty classes based on truncated CoT perfor-

mance: ***Simple***, ***Middle*** and ***Hard***. (1) ***Simple***: Queries where the model's full CoT reasoning produces the correct answer, and all truncated CoT sequences also yield correct answers. Here, minimal reasoning suffices; (2) ***Middle***: Truncated CoTs yield mixed results—some produce correct answers, while others fail. This means that the model requires sufficiently extended reasoning to succeed; (3) ***Hard***: Queries where the full CoT reasoning leads to an incorrect answer, and all truncated CoT sequences fail. These queries remain unresolved regardless of reasoning length.

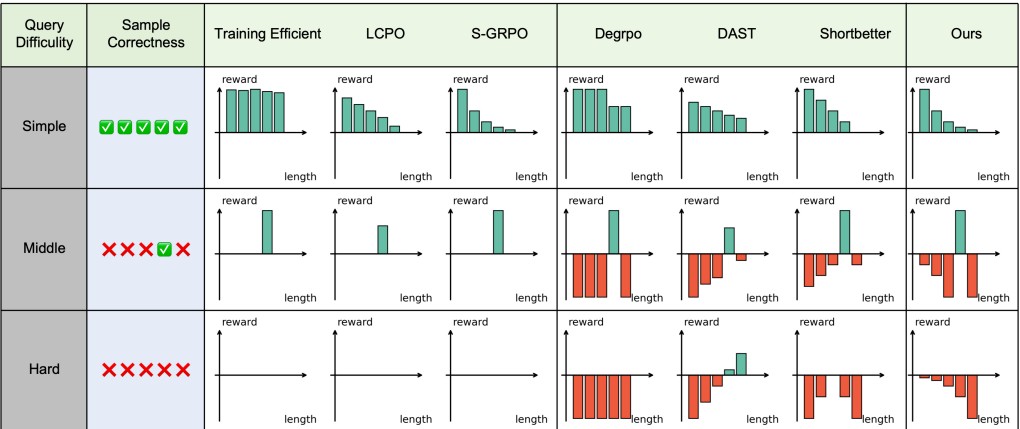

Figure 1: Comparison of reward strategies in recent RL methods.

Figure 1 compares reward strategies in recent RL methods. Training Efficient (Arora & Zanette, 2025) adopts a uniform strategy, granting comparable rewards to all positive samples and zero reward to negative samples. Conversely, LCPO (Aggarwal & Welleck, 2025) and S-GRPO (Dai et al., 2025b) differentially reward positive samples based on CoT length, assigning higher rewards to shorter successful reasoning paths. Degrpo (Fang et al., 2025) advances prior approaches by applying significant negative rewards to all negative samples. Recent state-of-the-art (SOTA) methods DAST (Shen et al., 2025) and Shorterbetter (Yi & Wang, 2025) employ similar strategies for positive samples as previous methods, while diverging in handling negative samples. DAST prioritizes answer correctness by progressively increasing rewards to encourage extended reasoning when answers are wrong. Shorterbetter instead rewards negative samples based on length proximity to the minimal sufficient CoT, incentivizing CoT lengths near optimal minima.

Despite their promise, current SOTA methods still face two critical limitations. First, for *Hard* queries, DAST promotes progressively longer CoTs while Shorterbetter converges to medium lengths. Given the low solve probability for these queries, both approaches waste substantial tokens generating futile reasoning paths. Second, for *Middle* queries, which require precise CoT lengths as slight deviations cause failure, both DAST and Shorterbetter encourage extending CoTs before reaching successful lengths. However, accumulating leading signals often causes models to overshoot the critical reasoning step, resulting in incorrect answers despite approaching correctness.

To address these limiations, we introduce RazorReward: a novel reward principle enforcing a sharp boundary between benefit and penalty. For *Hard* queries, it incentivizes minimal CoTs—solving directly without thinking or abstaining if unsolvable. For *Middle* queries, where correctness is hypersensitive to CoT length, RazorReward prioritizes exact attainment of the minimal sufficient reasoning length by heavily penalizing outputs that fall short or exceed it—eliminating wasteful "approaching correct" paths.

Accordingly, we propose RazorReward-RL, a novel reinforcement learning framework to mitigate *overthinking*. Unlike prior work (Dai et al., 2025b) that relies on arbitrary segmentation, our method partitions CoTs

into structural reasoning blocks, enabling construction of semantically coherent positive and negative samples for RL training. We further design a reward function that grants higher rewards to shorter, correct reasoning, imposes increasing penalties on longer, incorrect reasoning, and severely penalizes failures near the optimal length. This approach forces precise calibration of reasoning steps, eliminating token waste on "approaching correct" reasoning. Evaluation across six mathematical reasoning benchmarks (spanning varying difficulty levels) using DeepSeek-R1-Distill-Qwen-7B/1.5B backbones demonstrates that RazorReward-RL improves accuracy by 8.3%–9.3% while reducing average token consumption by 38.4%–43.8%, achieving a superior accuracy-efficiency trade-off over prior methods.

## 2 RELATED WORK

**Training-Free Methods** These methods mitigate the overthinking problem without LLMs fine-tuning. For instance, Deer (Dai et al., 2025b) dynamically halts generation by analyzing next-token logits; Concise (Qiao et al., 2025) makes stopping decisions based on the model's confidence in intermediate answers; and Dynasor-CoT (Yang et al., 2025b) dynamically stops inference by monitoring certainty. Differently, RouteLLM (Ong et al., 2025) employs routing strategies to assign queries to either strong or weak models, while ThinkSwitch (Liang et al., 2025) introduces a lightweight regressor to switch between different reasoning modes.

**Supervised Fine-Tuning Methods** LS-Mixture (Yu et al., 2025a) constructs a dataset that combines long CoT examples with structurally compressed short CoT examples for model fine-tuning. Similarly, Z1 (Yu et al., 2025b) creates a dataset containing both short and long CoT variants while removing explicit thinking markers. While these supervised fine-tuning (SFT) methods enable models to autonomously select reasoning modes, it places high demands on the quality of training data, requiring substantial manual effort.

**Reinforcement Learning Methods** AdaptThink (Zhang et al., 2025) dynamically selects between long and short reasoning modes, while Adacot (Lou et al., 2025) penalizes incorrect mode choices to prevent over/under-thinking. De-GRPO (Fang et al., 2025) prioritizes concise and correct reasoning through targeted rewards to enhance efficiency-accuracy tradeoffs. Other methods adopt explicit length-based rewards: Training Efficient (Arora & Zanette, 2025) and LCPO (Aggarwal & Welleck, 2025) reward correct answers shorter than an average/preset length. DAST (Shen et al., 2025) adjusts reasoning length by question difficulty, encouraging longer chains for incorrect samples, whereas Shortbetter (Yi & Wang, 2025) incentivizes lengths near the first correct answer. S-GRPO (Dai et al., 2025b) introduces serial sampling with early-stopped inference paths to construct sampling groups.

Our method diverges from prior RL approaches in two key ways. First, instead of arbitrary segmentation to generate RL samples, we segment using inherent phrase boundaries. Second, while existing methods incrementally reward longer CoTs for incorrect answers—risking wasted tokens on futile paths—we employ a razor-like reward mechanism. Chains leading towards or including key steps without the correct answer face severe penalties. This forces the model to identify essential steps efficiently, minimizing wasted effort on near-correct reasoning.

## 3 METHOD

This section introduces RazorReward-RL, a two-stage approach. In the first stage, CoT sequences are truncated and appended with an early-stopping prompt to generate positive and negative samples. In the second stage, we construct the RazorReward function and then perform RL to optimize the model. The overview of RazorReward-RL is presented in Figure 2.

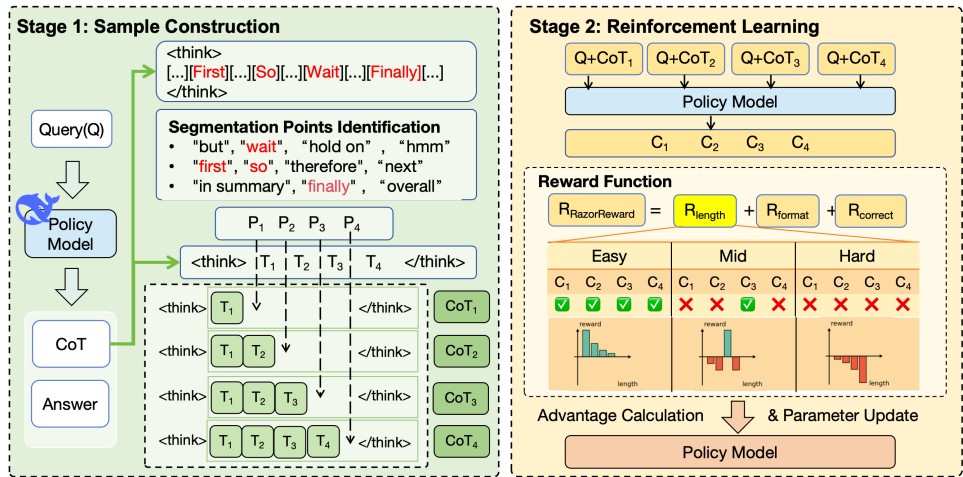

Figure 2: Overview of our RazorReward-RL framework.

## 3.1 SAMPLE CONSTRUCTION

Given an input query $q$, the policy model $\pi_\theta$ first generates a complete chain of thought and a final answer, i.e., $\langle s_0, c_0 \rangle = \pi_\theta(q)$, where $s_0 = [t_1, t_2, \ldots, t_n]$ is the token sequence representing the chain of thought, consisting of $n$ reasoning tokens, and $c_0$ is the model's final answer derived from the chain of thought.

### 3.1.1 SEGMENTATION POINTS IDENTIFICATION

A chain of thought typically consists of a sequence of logically coherent and atomic steps. Segmentation points should align with natural transitions or stage boundaries in the reasoning process. To identify such segmentation points, our approach leverages a predefined lexicon of structural phrases. This lexicon serves as a reservoir of markers indicative of reasoning boundaries. Segmentation occurs by matching entries from this lexicon against the CoT sequence. For efficient and comprehensive matching across the lexicon, we employ the Aho-Corasick multi-pattern string matching algorithm (Thorbecke et al., 2024). The lexicon includes a diverse range of markers indicating reasoning transitions, stepwise progression, hypothesis formation and summary statements. A comprehensive enumeration of the phrases is provided in the Appendix A.1.

### 3.1.2 EARLY-STOPPING GENERATION

To this end, we derive a list of segmentation points $P = [p_1, \cdots, p_m]$. Each point $p_i$ specifies a token index within the CoT sequence, and $m$ represents the total number of points. These points partition the CoT sequence $c_0$ into $m + 1$ distinct reasoning steps $\{T_1, \cdots, T_{m+1}\}$, where $T_i$ denotes the token sequence of the $i$-th reasoning step.

Given a segmentation position $p_i$, the original CoT sequence is truncated, forming the sequence $s_i = T_1 \oplus \cdots \oplus T_i$, where $i = 1, \ldots, m$, and $\oplus$ denotes sequence concatenation. The policy model $\pi_\theta$ is then queried with each truncated sequence $s_i$ alongside the original question $q$, generating a corresponding answer $c_i$ as $c_i = \pi_\theta(q \oplus s_i \oplus \mathcal{P}_{es})$, where $\mathcal{P}_{es}$ represents the token sequence of the early-stopping prompt (e.g., *"The thinking time is up, please output the answer according to the format requirements."*).

All generated answers, paired with their respective (original or truncated) CoT sequences, are aggregated into a serial training group $G$: $G = \{< c_1, s_1 >, \cdots, < c_m, s_m >, < c_0, s_0 >\}$. A sample $< c_i, s_i >$ is labeled as a positive example if $c_i$ is correct; otherwise, it is labeled as a negative example. This group $G$ facilitates direct comparison and targeted reward assignment across answers generated at different reasoning depths, providing diverse positive and negative samples crucial for enhancing reinforcement learning.

## 3.2 REINFORCEMENT LEARNING

Leveraging the samples within $G$, we build the RL framework for model optimization. This section details the proposed reward function, RazorReward, and the corresponding training objective.

### 3.2.1 REWARD FUNCTION

The RazorReward function scores samples in $G$, extending beyond conventional binary correctness to incorporate three components: correctness reward (i.e., $R_{\text{correct}}$ prioritizes answer accuracy), format reward (i.e., $R_{\text{format}}$ ensures structural compliance), and length reward (i.e., $R_{\text{length}}$ promotes reasoning efficiency). The total reward for a sample $x_i \in G$ is:

$$R_{\text{RazorReward}}(x_i) = R_{\text{correct}}(x_i) + R_{\text{format}}(x_i) + R_{\text{length}}(x_i). \tag{1}$$

**Correctness Reward**: A sample receives 2 points for a fully correct answer and 0 otherwise. This prioritizes accuracy over efficiency or early termination, aligning with standard RL practices. **Format Reward**: To ensure structured outputs for reliable evaluation, a sample must adhere to the format `<think>...</think> <answer>...</answer>`—requiring all tags to appear exactly once in the correct order. Compliance yields 1 point; non-compliance yields 0, eliminating ambiguity from inconsistent formatting. **Length Reward**: This component mitigates redundant reasoning via stepwise reward. Let $\text{Index}_{\text{right}}$ denote the index of the first correct sample in $G$ (or $n$ if none exists), and $\text{Num}^i_{\text{right}}$ the cumulative correct samples up to $x_i$. The reward $r_i$ for $x_i = < c_i, s_i >$ is:

$$r_i = \begin{cases} \dfrac{1}{2^{\text{Num}^i_{\text{right}} - 1}}, & \text{if } c_i \text{ is correct.} \\ -\dfrac{0.5}{2^{\text{Index}_{\text{right}} - i - 1}}, & \text{if } i < \text{Index}_{\text{right}} \text{ and } c_i \text{ is incorrect.} \\ -0.5, & \text{if } i > \text{Index}_{\text{right}} \text{ and } c_i \text{ is incorrect.} \end{cases} \tag{2}$$

The interpretation is as follows: (1) **Correct samples**: Rewards decay exponentially with each subsequent correct answer (e.g., 1 for the first, 0.5 for the second), incentivizing concise reasoning; (2) **Errors before first correct**: Penalties intensify near $\text{Index}_{\text{right}}$, discouraging mistakes close to correctness; (3) **Errors after first correct**: Fixed penalty of $-0.5$ suppresses any subsequent incorrect or redundant steps, once a correct sample has appeared. With such length reward, RazorReward jointly prioritizes the shortest correct CoT path, discourages excessively long reasoning when queries are unsolvable, and mitigates token wastage on near-correct reasoning through severe penalties on samples near the correct samples in $G$.

## 3.3 TRAINING OBJECTIVE

For a query $q$, consider a group of $m + 1$ samples $G = \{\langle c_1, s_1 \rangle, \ldots, \langle c_m, s_m \rangle, \langle c_0, s_0 \rangle\}$, each assigned a reward $r_i$ by the reward function. The advantage $\hat{A}_i$ for the sample $x_i \in G$ is computed as $\hat{A}_i = r_i - \frac{1}{m+1} \sum_{j=0}^{m} r_j$, where the denominator $m + 1$ is the group size. To maintain training stability, the computed advantage $\hat{A}_i$ is assigned uniformly to every token in the corresponding answer $c_i$, such that for any token

(e.g., index $t$) in $c_i$, we set $\hat{A}_{i,t} = \hat{A}_i$. The policy optimization employs a clipped surrogate objective:

$$
\mathcal{J}(\theta) = \mathbb{E}_{q \sim P(Q),\, \{c_i\}_{i=1}^{|G|} \sim \pi_{\theta_{\text{old}}}(C|q)}
$$

$$
\left[ \frac{1}{|G|} \sum_{i=1}^{|G|} \frac{1}{|c_i|} \sum_{t=1}^{|c_i|} \left\{ \min\left( \frac{\pi_\theta^{i,t}}{\pi_{\theta_{\text{old}}}^{i,t}} \hat{A}_i,\ \text{clip}\left( \frac{\pi_\theta^{i,t}}{\pi_{\theta_{\text{old}}}^{i,t}}, 1-\epsilon, 1+\epsilon \right) \hat{A}_i \right) \right\} \right], \tag{3}
$$

Here, $q$ represents the input query. $\pi_\theta$ and $\pi_{\theta_{\text{old}}}$ are the current and reference policy parameters before optimization, respectively. The expression $\pi^{i,t} = \pi(c_{i,t} \mid q', c_{i,<t})$ designates the probability of generating token $c_{i,t}$ at position $t$ in answer $c_i$, given prompt $q'$ and preceding tokens $c_{i,<t}$, where $q' = q \oplus s_i \oplus \mathcal{P}_{\text{es}}$ denotes the prompt constructed by appending the CoT sequence $s_i$ and early-stopping prompt $\mathcal{P}_{\text{es}}$ to the input query. The hyperparameter $\epsilon$ clips the importance sampling ratio to ensure stable policy updates.

## 4 EXPERIMENT

### 4.1 SETUP

**Datasets** Following prior works (Dai et al., 2025b), we constructed DeepMath-30K Balanced, a difficulty-balanced training set, by sampling problems from the DeepMath-103K dataset (He et al., 2025) (covering grades 5-10 mathematics). After pre-processing, we obtain a training set of 30190 samples.[1] We evaluate our model on six math and science reasoning benchmarks: GSM8K (Cobbe et al., 2021) test set (1319 grade school math problems),MATH500 (Hendrycks et al., 2021b)(500 high-school competition problems), AIME 2024(MAA Committees, 2024) and AIME 2025(MAA Committees, 2025) (30 Olympiad-level problems each year), AMC 2023 (AI-MO, 2024) (40 high school competition problems), and GPQA_D (Rein et al., 2023) (198 graduate-level science questions).

**Baselines** The RL methods include ShorterBetter (Yi & Wang, 2025), DAST (Shen et al., 2025), L1-Max (Aggarwal & Welleck, 2025) and AdaptThink (Zhang et al., 2025). Turning to training-free methods, we evaluate against DEER (Dai et al., 2025b). For detailed descriptions, please refer to the Related Works section. Additionally, we include the Vanilla model for comparison. This model directly uses the backbone for inference without any fine-tuning.

**Implementation Details** For each query, we constrain the number of segmentation points to 8. Key hyperparameters during training include a 2048-token response length, batch size of 128, and learning rate of 1e-6. The 2048-token limit aligns with production systems like Baidu Search (Baidu, 2025) and Doubao (ByteDance, 2025), where most user queries need responses shorter than 2000 tokens (Lou et al., 2025). The selected response limit facilitates adaptation to production environments like search engines in the future. All baselines are rigorously reproduced using their officially released model weights. Following prior work (Yi & Wang, 2025), Deepseek-R1-Distill-Qwen-7B and 1.5B are used as backbones.

**Evaluation Protocol** We evaluate model performance using three primary metrics: **Accuracy (Acc↑)**, **Output Length (Tok↓)** and **Accuracy-Efficiency Score (AES↑)**. Output Length measures the average number of tokens generated per sample, where lower values reflect more concise responses. The Accuracy-Efficiency Score (AES), introduced to jointly evaluate output brevity and accuracy preservation (Luo et al., 2025). A higher AES indicates better efficiency with minimal or no loss in correctness.[2]

---

[1] Due to space limitations, pre-processing details are provided in the Appendix A.3

[2] Due to space limitations, the details of AES are provided in the Appendix A.4

Table 1: Experimental results on six mathematical and scientific reasoning benchmarks. Best and second-best performance for both Acc and AES are marked in **bold** and underline, respectively.

| Method | GSM8K | | AIME24 | | AMC23 | | MATH-500 | | GPQA_D | | AIME25 | | Summary | | |
|---|---|---|---|---|---|---|---|---|---|---|---|---|---|---|---|
| | Acc↑ | Tok↓ | Acc↑ | Tok↓ | Acc↑ | Tok↓ | Acc↑ | Tok↓ | Acc↑ | Tok↓ | Acc↑ | Tok↓ | Acc↑ | Tok↓ | AES↑ |
| DeepSeek-R1-Distill-Qwen-7B | | | | | | | | | | | | | | | |
| Vanilla | 77.5 | 406 | 28.5 | 3735 | 59.5 | 2739 | 73.5 | 1580 | 21.1 | 2604 | 22.9 | 3700 | 47.17% | 2461 | |
| ShorterBetter | 86.6 | 109 | 31.0 | 2299 | 68.6 | 1000 | 76.6 | 492 | 36.9 | 780 | 20.0 | 2030 | 53.28%(+6.11%) | 1118 (-54.6%) | 0.934 |
| DAST | **91.3** | 628 | 25.6 | 3754 | 66.1 | 2709 | 80.5 | 1802 | 20.6 | 3061 | 23.8 | 3717 | 51.32%(+4.15%) | 2612 (+6.1%) | 0.203 |
| L1-Max | 57.4 | 515 | 30.8 | 2075 | 68.4 | 1434 | 70.1 | 1059 | 30.6 | 1152 | 18.8 | 1873 | 46.02%(-1.15%) | 1351(-45.1%) | 0.378 |
| AdaptThink | 83.2 | 365 | 33.3 | 3454 | 72.8 | 2240 | **81.7** | 1296 | 27.7 | 2579 | **26.0** | 3481 | 54.12%(+6.95%) | 2236(-9.1%) | 0.533 |
| Deer | 85.7 | 382 | 20.0 | 1969 | 67.5 | 1261 | 70.0 | 772 | 29.3 | 1552 | 16.7 | 1930 | 48.20%(+1.03%) | 1311(-46.7%) | 0.533 |
| RazorReward-RL | 89.2 | 447 | **33.3** | 2490 | **74.7** | 1656 | 81.2 | 1088 | **39.0** | 1243 | 21.3 | 2168 | **56.45%(+9.28%)** | 1515(-38.4%) | **0.975** |
| DeepSeek-R1-Distill-Qwen-1.5B | | | | | | | | | | | | | | | |
| Vanilla | 73.8 | 442 | 14.6 | 3952 | 48.3 | 3021 | 66.6 | 2063 | 8.1 | 2790 | 13.3 | 3912 | 37.45% | 2697 | |
| ShorterBetter | 78.0 | 362 | 15.2 | 2824 | 56.7 | 1594 | 73.0 | 992 | 22.9 | 1149 | 13.3 | 2653 | 43.18%(+5.70%) | 1596(-40.8%) | 0.867 |
| L1-Max | 73.0 | 1480 | **22.7** | 2847 | **69.2** | 2777 | **76.5** | 2408 | 30.2 | 1166 | **19.2** | 2796 | 48.47%(+11.0%) | 2246(-16.7%) | 1.050 |
| AdaptThink | **81.8** | 303 | 16.5 | 2012 | 55.5 | 1279 | 74.7 | 782 | 15.2 | 1667 | 12.3 | 1861 | 42.67%(+5.20%) | 1317(-51.2%) | 0.930 |
| Deer | 59.5 | 329 | 3.3 | 1948 | 25.0 | 1448 | 37.6 | 846 | 2.0 | 1827 | 3.3 | 1948 | 21.78%(-15.7%) | 1391(-48.4%) | -1.608 |
| RazorReward-RL | 80.9 | 383 | 17.9 | 2684 | 59.4 | 1661 | 71.2 | 1104 | **30.4** | 1037 | 14.4 | 2222 | 45.7% (+8.25%) | 1515 (-43.8%) | **1.099** |

## 4.2 MAIN RESULTS

Table 1 compares our method with recent baselines across six mathematical and scientific reasoning datasets and two LLM backbones. First, RazorReward-RL consistently outperforms the Vanilla model in accuracy across most datasets and both LLM backbones. Using the 7B backbone, RazorReward-RL achieves an average accuracy of 56.45% – a 9.28 percentage-point improvement over Vanilla – while reducing token consumption by approximately 38.4%. Similarly, on the 1.5B backbone, RazorReward-RL delivers an 8.25% accuracy gain alongside a 43.8% reduction in reasoning sequence length. These results collectively demonstrate the effectiveness and generalizability of our approach across diverse datasets and model scales.

Second, against recent baselines, RazorReward-RL achieves the best average accuracy on the 7B backbone and the second-best on the 1.5B backbone. Crucially, it attains the best AES across both backbones, demonstrating superior overall trade-offs. While ShorterBetter ranks second in AES on 7B, RazorReward-RL outperforms it on the 1.5B backbone in both accuracy and token reduction, leading to significantly higher AES. On 1.5B, L1-Max achieves higher accuracy but ranks second in AES; this reverses on the 7B backbone, where RazorReward-RL exceeds L1-Max by 10.43% in accuracy and significantly in AES. These results underscore RazorReward-RL's superior generalizability.

## 4.3 ABLATION STUDY

This section presents an ablation study to quantify the contributions of our framework across six benchmarks and two backbone models. The compared models are: (1) **Vanilla**: This baseline utilizes the backbone for inference without fine-tuning; (2) **Basic RL**: This model optimizes the Vanilla baseline using the basic RL framework outlined in S-GPRO (Yi & Wang, 2025). It serves as our primary ablation point due to its similar pipeline as ours; (3) **Our RazorReward**: This model optimizes the Vanilla baseline using our full RL framework. Specifically, it replaces the random CoT segmentation used in Basic RL with our structural segmentation and substitutes the reward function with RazorReward. The results are shown in Table 2.

The Basic RL model significantly outperforms the Vanilla baseline across most datasets and backbones, achieving an average token reduction rate ranging from 38% to 52%. This validates the effectiveness of the basic framework in optimizing the backbone performance. Crucially, our RazorReward model consistently

Table 2: Ablation study on six mathematical reasoning benchmarks.

| Method | GSM8K | | AIME24 | | AMC23 | | MATH-500 | | GPQA_D | | AIME25 | | Summary | | |
|---|---|---|---|---|---|---|---|---|---|---|---|---|---|---|---|
| | Acc↑ | Tok↓ | Acc↑ | Tok↓ | Acc↑ | Tok↓ | Acc↑ | Tok↓ | Acc↑ | Tok↓ | Acc↑ | Tok↓ | Acc↑ | Tok↓ | AES↑ |
| DeepSeek-R1-Distill-Qwen-7B | | | | | | | | | | | | | | | |
| Vanilla | 77.5 | 406 | 28.5 | 3735 | 59.5 | 2739 | 73.5 | 1580 | 21.1 | 2604 | 22.9 | 3700 | 47.17% | 2461 | |
| w/ Basic RL | 87.5 | 426 | 25.8 | 2642 | 71.4 | 1498 | 79.6 | 980 | 33.1 | 1273 | 21.0 | 2254 | 53.08%(+5.91%) | 1512(-38.6%) | 0.759 |
| w/ Our RazorReward | **89.2** | 447 | **33.3** | 2490 | **74.7** | 1656 | **81.2** | 1088 | **39.0** | 1243 | 21.3 | 2168 | **56.45%**(+9.28%) | 1515(-38.4%) | **0.972** |
| DeepSeek-R1-Distill-Qwen-1.5B | | | | | | | | | | | | | | | |
| Vanilla | 73.8 | 442 | 14.6 | 3952 | 48.3 | 3021 | 66.6 | 2063 | 8.1 | 2790 | 13.3 | 3911 | 37.46% | 2696 | |
| w/ Basic RL | 77.3 | 315 | 14.4 | 2296 | 54.7 | 1369 | 70.6 | 912 | 26.8 | 859 | 12.7 | 1954 | 42.74%(+5.28%) | 1284(-52.4%) | 0.947 |
| w/ Our RazorReward | **80.9** | 383 | **17.9** | 2684 | **59.4** | 1661 | **71.2** | 1104 | **30.4** | 1037 | 14.4 | 2222 | **45.71%**(+8.25%) | 1515(-43.8%) | **1.099** |

surpasses the Basic RL model on accuracy across all datasets and backbones. While the average token reduction achieved by RazorReward is lower in certain cases, it delivers a consistently higher AES. This demonstrates RazorReward's superior ability to balance model accuracy against inference efficiency.

### 4.3.1 FURTHER ANALYSIS

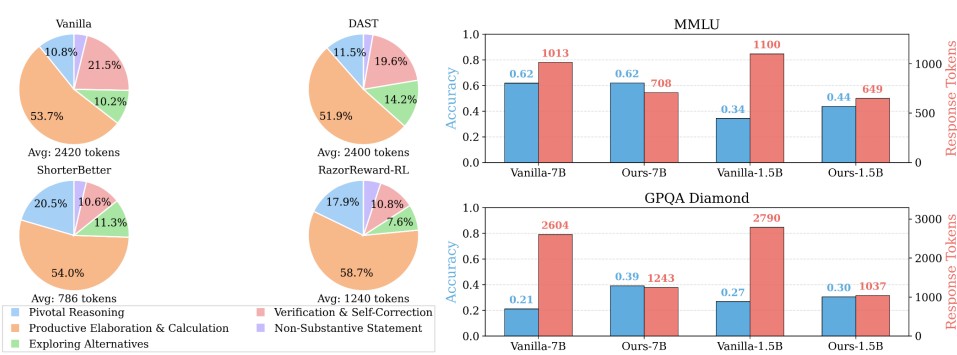

Figure 3: Left: Breakdown of reasoning step categories and path lengths across different models. Right: Out-of-distribution (OOD) generalization performance: accuracy and token reduction under 1.5B/7B backbones.

**Reasoning Quality Analysis** We evaluate reasoning quality by categorizing CoT paths (using Qwen3-235B-A22B (Yang et al., 2025a); prompts in Appendix A.2) into five mutually exclusive classes (*Pivotal Reasoning*, *Productive Elaboration & Calculation*, *Exploring Alternatives*, *Verification & Self-Correction*, *Non-substantive Statements*), assessing category proportions at the token level. Comparisons against Vanilla and RL baselines (DAST, Shortbetter) on the 7B backbone are shown in Figure 3 (Left).

First, RazorReward-RL achieves the highest share of *Pivotal Reasoning* + *Productive Elaboration & Calculation* (76.6%), outperforming all compared baselines. Second, RazorReward-RL minimizes *Exploring Alternatives* + *Verification & Self-Correction* (18.4%), indicating a more direct process (Shortbetter shows more exploration). Third, RazorReward-RL maintains reasoning quality with a significantly shorter average path length (1240 tokens) than DAST and Vanilla. Overall, RazorReward-RL achieves more focused, efficient, and concise reasoning than all baselines.

**Performance under Different Difficulty Levels** This section compares model performance across three query difficulty levels (Easy, Middle, and Hard). We randomly selected 250 queries from the Math500 dataset with an Easy:Middle:Hard ratio of 1:2:2. Corresponding positive and negative samples were generated using the DeepSeek-R1-Distill-Qwen-7B model. Results are presented in Table 3.

At Easy level, all models achieved accuracy higher than 90%, with RazorReward-RL performing best (98.75%). This exceeds DAST by 0.25% and ShorterBetter by 5.25%. Although RazorReward-RL and DAST demonstrate comparable accuracy, RazorReward-RL achieves significantly greater token reduction.

At Middle level, DAST achieved the highest accuracy (95.25%), while RazorReward-RL attained comparable accuracy (94.88%) with greater reasoning efficiency. RazorReward-RL significantly reduced token usage, especially on incorrect responses, and achieved the lowest Error/Correct Ratio, demonstrating its effectiveness in minimizing exploration of near-correct paths and focusing tokens on correct reasoning, consistent with our design principle to penalize deceptive reasoning and maintain a controllable process.

At Hard level, DAST maintained the highest accuracy (76.38%) but exhibited excessive reasoning lengths. RazorReward-RL balanced accuracy (74.88%) and efficiency effectively, while ShorterBetter's aggressive token reduction strategy resulted in significant accuracy loss (63.25%). With an optimal Error/Correct Ratio of 1.68, RazorReward-RL demonstrates effective balancing—avoiding overthinking while preserving problem-solving capability on challenging problems.

Table 3: Performance comparison across difficulty levels: Easy (purple), Middle (orange), Hard (green). **Error/Correct Ratio = Incorrect Tok / Correct Tok**.

| Model | Acc↑ | Avg Tok↓ | Correct Tok↓ | Incorrect Tok↓ | Error/Correct Ratio↓ |
|---|---|---|---|---|---|
| Vanilla | 92.50 | 816.49 | 847.11 | 438.83 | 0.52 |
| **RazorReward-RL** | **98.75** | 668.82 | 665.28 | 948.80 | 1.43 |
| DAST | 98.50 | 1102.07 | 1056.48 | 4096.00 | 3.88 |
| ShorterBetter | 93.50 | 154.93 | 145.89 | 285.08 | 1.95 |
| Vanilla | 88.88 | 1129.93 | 1004.56 | 2131.48 | 2.12 |
| RazorReward-RL | 94.88 | 884.26 | 846.66 | 1580.37 | 1.87 |
| **DAST** | **95.25** | 1413.92 | 1284.28 | 4013.55 | 3.13 |
| ShorterBetter | 90.88 | 348.51 | 292.03 | 910.97 | 3.12 |
| Vanilla | 68.38 | 1435.01 | 1168.48 | 2011.27 | 1.72 |
| RazorReward-RL | 74.88 | 1066.36 | 910.83 | 1529.85 | 1.68 |
| **DAST** | **76.38** | 1707.25 | 1447.17 | 2548.05 | 1.76 |
| ShorterBetter | 63.25 | 446.53 | 316.49 | 670.33 | 2.12 |

**Generalizability in Out-of-Distribution (OOD) Scenarios**   To further evaluate our approach under OOD conditions, we test on MMLU (Hendrycks et al., 2021a) and GPQA Diamond. Both benchmarks diverge significantly from our math-focused training corpus: MMLU covers 14K multi-choice questions across 57 diverse domains, while GPQA Diamond comprises scientific question-answering tasks.

As shown in Figure 3 (Right), our models exhibit robust generalization: (1) on MMLU, using both 7B and 1.5B backbones, our approach achieves comparable or higher accuracy than the Vanilla model while reducing response tokens by 30–40%; (2) On GPQA Diamond, our models improve accuracy by 18% and substantially shorten response length, demonstrating enhanced efficiency and adaptability in OOD scenarios.

## 5 CONCLUSION

This study introduced RazorReward-RL, a novel RL framework addressing LLM *overthinking*. By segmenting CoTs into structural reasoning blocks, RazorReward-RL enables the construction of semantically coherent training samples. Its core RazorReward function imposes large penalization on both under- and over-reasoning relative to the minimal sufficient CoT. This allows precise calibration of reasoning steps, effectively reducing model's token waste on the failed reasoning near the optimal length. Extensive experiments across six benchmarks demonstrate RazorReward-RL's superior accuracy-efficiency trade-off. Further analyses, including reasoning quality assessment, performance on queries of varying difficulty and OOD experiments, consistently validate the framework's efficacy.

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

## A  APPENDIX

### A.1  ENUMERATION OF REASONING MARKERS

To systematically identify and analyze reasoning processes in text, we curated a comprehensive set of structured linguistic markers. These phrases serve as signals for reasoning transitions, stepwise progressions, analytical constructs, and conclusion statements. For clarity and reproducibility, we categorize the markers as follows:

1. **Transitions and Self-Verification**
   These phrases signal a shift in reasoning, pausing for self-checks, or considering alternatives. Examples include: *but*, *wait*, *hold on*, *hmm*, *let's check*, *let me double-check*, *alternatively*

2. **Sequential Progressions**
   These markers indicate the logical progression of steps or ideas: *so*, *after that*, *next*, *then*, *therefore*, *given that*, *together*, *in total*, *thus*, *alright*, *finally*, *first*, *now*

3. **Step Indicators**
   Explicit numbering or phrasing denoting individual steps: *step 1*, *step 2*, *1.*, *2.*, *to find*

4. **Analytical and Hypothetical Constructs**
   Phrases that introduce analysis, hypotheses, or reference information: *let's analyze*, *let's consider*, *suppose*, *assume that*, *notice that*, *recall that*

5. **Summarization and Conclusion**
   Markers signaling the end of reasoning or summarization: *in summary*, *in conclusion*, *overall*

### A.2  PROMPT TEMPLATE FOR REASONING QUALITY ANALYSIS

---

**Prompt for Reasoning Quality Analysis**

You are a reasoning trace analyst. Your task is to categorize each line (separated by a newline, except for equations, which should not be considered standalone lines) in a given reasoning trace according to its role in the reasoning process. The objective is to understand how different components of the model's reasoning contribute to the final answer.

You will receive a complete reasoning trace ending with a final answer after the `</think>` tag. You must:

- Only analyze the content before the `</think>` tag.

- Split the reasoning trace into individual lines. Multiple sentences may appear in a single line; a line ends at a newline character.

- Assign exactly one label to each line from the following mutually exclusive categories, based on its primary function in context.

**Categories:**

- **Pivotal Reasoning:** Steps that directly correspond to specific parts of the final summary or solution (as shown after `</think>`). These include essential equations, key variable assignments, or critical conclusions explicitly present in the summarized answer.

- **Productive Elaboration & Calculation:** Necessary calculations, logical deductions, planning, or explanations that support a pivotal step but are not themselves restated in the final summary.

---

- **Exploring Alternatives:** Attempts to try different approaches, propose hypotheses, or check other methods that are ultimately not used in the final solution.
- **Verification & Self-Correction:** Sentences in which the model checks, verifies, or corrects earlier results to catch errors or reconsider its approach.
- **Non-Substantive Statement:** Redundant comments, conversational fillers, or trivial rephrasings that do not advance the solution or add meaningful structure.

**Additional Instructions:**

- Stop processing as soon as the `</think>` tag is encountered. Do not categorize anything beyond it. The content after `</think>` serves as the reference for "Pivotal Reasoning."
- If a line repeats or paraphrases an earlier line without adding new value or serving a clear structural purpose (such as summarizing inputs before a new calculation phase), categorize the repeated instance as "Verification & Self-Correction," even if the original served a different purpose.
- If a line could arguably fit more than one category, choose the one that best describes its primary function or most specific contribution in context. For example, a calculation that corrects a previous error is "Verification & Self-Correction" rather than "Exploring Alternatives."
- Do not infer the logical correctness of the reasoning or the final answer. The categorization concerns the structure and the perceived purpose of each statement within the model's reasoning process.
- Treat each line independently, but utilize surrounding context (preceding and succeeding lines) for understanding its function, especially in identifying repetitions, planning statements, or logical flow.
- If the reasoning trace starts with the `</think>` tag, return an empty list.

**Output Format:**

- Return your output as a JSON array of objects.
- Each object should have: `"text"`: the full original line (string), and `"label"`: one of the five category names above (string).

**Example Output:**

```
[{"text": "The problem asks for the speed of the train.",
"label": "Exploring Alternatives"},
 {"text": "We set up the equation: d = s * t",
 "label": "Pivotal Reasoning"}]
```

Return only the structured JSON list, without any extra commentary or explanation.

**Reasoning trace to analyze:**

```
{response}
```

## A.3 TRAINING DATASET CONSTRUCTION DETAILS

We construct the training set by stratified sampling from DeepMath-103K, a large-scale math problem dataset spanning grades 5-10, with difficulty levels annotated from 3 to 9 via multi-round AI assessment. To ensure training stability and representativeness, we first exclude samples with unclassifiable answers (non-numeric/yes-no), then compute the joint distribution of difficulty and topic. We sample 30,190 problems

such that the marginal and conditional distributions over both difficulty and topic closely match those of the filtered source data.

To further improve training stability and prevent sudden increases in problem difficulty within batch, we reorder the sampled dataset so that the sum of difficulties in each batch (batch size 32 or 128) remains nearly constant across all batches. This is achieved via a greedy assignment ensuring that every batch's average difficulty is approximately 6.0, with minimal variance. Empirically, this leads to consistent learning dynamics and avoids training collapse due to abrupt difficulty spikes.

Sampling and balancing are implemented in Python, following a two-step process: (1) stratified sampling by difficulty and topic, (2) batch-wise reordering for difficulty balancing.

### A.4 ACCURACY-EFFICIENCY SCORE

The Accuracy-Efficiency (AE) Score, as proposed by Luo et al. (2025), offers a metric for assessing whether a model can enhance inference efficiency—specifically, by shortening output length—while maintaining accuracy. The AE Score is formally defined by the following piecewise equation:

$$\text{AES} = \begin{cases} \Delta\text{Length} + \eta \cdot |\Delta\text{Acc}|, & \text{if } \Delta\text{Acc} \geq 0 \\ \phi \cdot \Delta\text{Length} - \theta \cdot |\Delta\text{Acc}|, & \text{if } \Delta\text{Acc} < 0 \end{cases} \tag{4}$$

where $\Delta\text{Length}$ and $\Delta\text{Acc}$ denote the percentage reductions in output length and accuracy relative to the base model. Following prior work (Luo et al., 2025), we set $\phi = 1$, $\eta = 3$, and $\theta = 5$. A higher AES indicates better efficiency with minimal or no loss in correctness.

