# OpenReview forum: "Surgical Trimming: Minimal Sufficient Chain of Thought with RazorReward-RL"
_ICLR.cc/2026/Conference — ICLR 2026 Conference Withdrawn Submission_

### Official Review · Reviewer_SdpA · 2025-10-28

**Soundness:** 3
**Presentation:** 2
**Contribution:** 2
**Rating:** 4
**Confidence:** 3

**Summary:**

This paper proposes RazorReward-RL, a reinforcement learning framework designed to reduce redundant reasoning in large language models by identifying and enforcing the minimal sufficient chain of thought (CoT). The authors introduce a novel RazorReward function that imposes sharp penalties for both under- and over-reasoning relative to the optimal CoT length, improving reasoning efficiency without sacrificing accuracy. Through structural segmentation of CoT into semantic blocks, the method achieves more precise reward allocation. Experiments across six reasoning benchmarks show consistent gains in accuracy and substantial reductions in token usage.

**Strengths:**

The proposed method itself is coherent and easy to follow. The authors conduct experiments on several benchmarks to validate the effectiveness of the proposed method.

**Weaknesses:**

While RazorReward-RL presents a promising framework for mitigating overthinking in reasoning models, several weaknesses limit its theoretical depth, methodological generality, and scalability.

**(1) Incremental contribution.** Although the paper introduces a new reward shaping function, the innovation is largely an extension of existing RL-based CoT optimization methods such as DAST and ShorterBetter. The central idea—penalizing both under- and over-reasoning—is conceptually intuitive and has been partially explored in previous work. The main novelty lies in the “razor-like” sharpness of the penalty, which, while effective, may not constitute a fundamentally new learning paradigm.

**(2) Lack of theoretical justification.** The paper provides no formal theoretical analysis explaining why the minimal sufficient CoT principle should universally optimize reasoning efficiency and correctness. There is no proof or theoretical model showing how reward sharpness affects convergence stability or sample efficiency in RL training. As a result, the work remains primarily empirical, lacking a solid mathematical foundation to justify its design choices.

**(3) Heavy reliance on handcrafted components.** The segmentation lexicon for identifying reasoning blocks and the rule-based structure of the RazorReward function introduce hand-engineered bias. This reliance on manually defined linguistic markers may limit adaptability to other languages, domains, or reasoning styles. Moreover, small errors in segmentation could misassign rewards, leading to training instability.

**(4) Limited scalability and model size.** Experiments are conducted only on the 1.5B and 7B DeepSeek-R1-Distill-Qwen backbones. These are relatively small models compared to modern reasoning LLMs (e.g., 30B–70B). It remains unclear whether the proposed framework maintains its benefits or efficiency when scaled up. Larger-scale validation would strengthen the generality claim.

**(5) Narrow domain evaluation.** The benchmarks focus almost exclusively on mathematical and scientific reasoning. It is uncertain whether the same reward dynamics hold for open-domain, commonsense, or multimodal reasoning, where the notion of “minimal sufficient reasoning” is less clearly defined.

**Questions:**

1.	How sensitive is performance to the segmentation lexicon and its coverage across reasoning styles or languages?
2.	Can the approach be extended to open-ended or dialogue reasoning tasks, where “minimal sufficient” reasoning is ill-defined?
3.	How stable is training when applied to larger models (e.g., ≥30B parameters)?

---

> ### Author Response · Authors · 2025-11-28
> **Response to Reviewer SdpA (Part 1)**
>
> > 1. Although the paper introduces a new reward shaping function, the innovation is largely an extension of existing RL-based CoT optimization methods such as DAST and ShorterBetter. The central idea—penalizing both under- and over-reasoning—is conceptually intuitive and has been partially explored in previous work. The main novelty lies in the “razor-like” sharpness of the penalty, which, while effective, may not constitute a fundamentally new learning paradigm.
>
> Thanks for your valuable feedback.
>
>
> We acknowledge that RazorReward-RL shares foundations with existing RL approaches, like DAST and ShorterBetter. However, **the core differentiator among these methods lies the in reward strategies—the most crucial and fundamental component of RL-based models**. Our contribution centers on advancing this aspect through novel insights and distinctive methods, which we believe merits recognition.
>
>
> **Concretely, RazorReward-RL advances prior works in two key aspects:**
>
>
> - First, unlike prior approaches, RazorReward-RL employs a dual penalty mechanism. It not only penalizes under-reasoning but also imposes a strong penalty on over-reasoning near correctness – a common yet frequently overlooked scenario where reasoning flips to incorrectness near the right answer. To our knowledge, such razor-like penalties (under- and over-reasoning) remain unexplored in existing literature.
>
>
> - Second, we incorporate semantic truncation during the rollout phase. This ensures rewards are aligned with the LLM’s native reasoning structure, thus yielding a higher proportion of positive samples than random truncation. Consequently, during RL training, the model receives a more positive reward to improve its accuracy.
>
>
> Comprehensive experiments in our manuscript, supplemented by additional validation in our rebuttal, substantiate the method’s efficacy.
>
>
> Together, these aspects ensure both practical utility and novelty of our approach.
>
> ---
>
> > 2. The paper provides no formal theoretical analysis explaining why the minimal sufficient CoT principle should universally optimize reasoning efficiency and correctness. There is no proof or theoretical model showing how reward sharpness affects convergence stability or sample efficiency in RL training. As a result, the work remains primarily empirical, lacking a solid mathematical foundation to justify its design choices.
>
> Thank you for your valuable feedback.
>
>
> Our "Minimal Sufficient CoT" principle originates from the two fundamental objectives of Efficiency Reasoning: **correctness and efficiency.**
>
>
> - **1. "Sufficient" corresponds to reasoning correctness: A reasoning process is sufficient if the model can arrive at the correct answer through it.**
>
>
> - **2. "Minimal" corresponds to reasoning efficiency: Under the guarantee of sufficiency, the result should be achieved using as few tokens as possible.**
>
>
> While we acknowledge the need for greater theoretical grounding, our reward strategy finds its justification through comprehensive experiments:
>
>
> 3. In the **main experiments**, our method demonstrates consistent improvements over baselines on both accuracy and token consumption. The Accuracy-Efficiency Score (AES) confirms that our method achieves a better balance between correctness and conciseness.
>
>
> 4. In the **reasoning quality analysis (Section 4.3)**, we observed a 12.1% increase in the proportion of tokens allocated to the core reasoning steps, alongside a 13.3% decrease in tokens devoted to redundant exploration.
>
>
> 5. Furthermore, **generalization experiments (conducted across diverse benchmarks like MMLU, GPQA, and HotpotQA)** consistently show stable performance gains.
>
>
> 6. The **supplementary experiments we added during the rebuttal phase (including chunk-level evaluation, verification step analysis, reasoning experiments with contexts up to 16k tokens, and logical integrity assessment)** provide further multi-faceted validation.
> We see formalizing these empirical insights into theoretical models as a valuable direction for future work and appreciate your constructive feedback on this point.

---

> ### Author Response · Authors · 2025-11-28
> **Response to Reviewer SdpA (Part 2)**
>
> > Weekness  (3) The segmentation lexicon for identifying reasoning blocks and the rule-based structure of the RazorReward function introduce hand-engineered bias. This reliance on manually defined linguistic markers may limit adaptability to other languages, domains, or reasoning styles. Moreover, small errors in segmentation could misassign rewards, leading to training instability.
>
> > Question （1）How sensitive is performance to the segmentation lexicon and its coverage across reasoning styles or languages?
>
> Thank you for this insightful comment.
>
>
> Regarding the adaptability to other domains, we conducted additional testing on HotpotQA, a benchmark commonly used in search domains. Results from **HotpotQA** demonstrate RazorReward-RL's strong performance, **achieving 16.88% EM and 23.13% F1 score**, significantly outperforming baseline models like ShorterBetter (8.59%), AdaptThink (6.21%) and DAST (4.83%). These results together with that on **MMLU, GPQA datasets (Line 404 - 411)** indicate RazorReward-RL’s strong generalizability.
>
> **Table 1. Reasoning performance on the HotpotQA dataset (EM and F1 scores).**
> | Model              | EM   | F1   |
> |--------------------|----------|----------|
> | RazorReward-RL     | 0.1688   | 0.2313   |
> | ShorterBetter      | 0.0859   | 0.1381   |
> | AdaptThink         | 0.0621   | 0.0978   |
> | ShorterBetter      | 0.0483   | 0.0752   |
> | Vanilla            | 0.0309   | 0.0530   |
>
> Regarding the adaptability to other reasoning styles, we shift our backbone to the **Qwen2.5-1.5B-Ins model**. RazorReward-RL maintained strong performance, showing an accuracy improvement **from 24.75% to 28.93% while the average token length decreased from 768 to 685.** This reinforces our method's adaptability.
>
> **Table 2. Performance of RazorReward-RL on Qwen2.5-1.5B-Instruct model across various datasets (accuracy and token usage).**
> | Model            | aime_2024 Acc↑ | aime_2024 Tok↓ | aime_2025 Acc↑ | aime_2025 Tok↓ | amc2023 Acc↑ | amc2023 Tok↓ | gqpa_diamond Acc↑ | gqpa_diamond Tok↓ | gsm8k Acc↑ | gsm8k Tok↓ | math500 Acc↑ | math500 Tok↓ | avg_acc | avg_tok | Acc_delta | Tok_delta | AES   |
> |------------------|----------------|----------------|----------------|----------------|--------------|--------------|-------------------|-------------------|------------|------------|--------------|--------------|---------|---------|-----------|-----------|-------|
> | Vanilla   | 0.033          | 1499           | 0.000          | 1299           | 0.225        | 839          | 0.152             | 762               | 0.613      | 279        | 0.462        | 662          | 0.248   | 768     | 0.00%     | 0.00%     | 0.000 |
> | RazorReward-RL | 0.033      | 933            | 0.033          | 1010           | 0.200        | 754          | 0.288             | 654               | 0.739      | 265        | 0.442        | 497          | 0.289   | 685     | 4.18%     | 10.78%    | 0.614 |
>
> Regarding the sensitivity of the segmentation lexicon, we experimented with two strategies to modify our lexicon set: (1) Synonym substitution within the lexicon. (2) Lexicon expansion. Both strategies are implemented through Qwen3-Max. The results are presented in the table below:
>
> **Table 3: Sensitivity analysis results for lexicon content variation.**
>
> | Lexicon Mode         | Acc↑   | Tok↓     |
> |----------------------|--------|----------|
> | Our Lexicon          | 0.464  | 7,334.8  |
> | Extended Lexicon     | 0.471  | 7,372.3  |
> | Synonym Replacement  | 0.470  | 7,112.2  |
>
> **We find that performance exhibits low sensitivity to adjustments.** Synonym substitutions and lexicon expansions yield only marginal accuracy gains of 0.64% and 0.07% respectively (in average), while increasing response length by 10%.
>
>
> **In summary, these results demonstrate our segmentation strategy’s generalizability across diverse tasks and reasoning styles, with lexicon perturbation showing minimal impact on performance.**
>
>
> Nevertheless, we acknowledge the limited adaptability to other languages since our lexicon is purely English, focusing on the widely used benchmarks for this problem. We plan to address other languages in future work.

---

> ### Author Response · Authors · 2025-11-28
> **Response to Reviewer SdpA (Part 3)**
>
> > Weekness  (4)  Experiments are conducted only on the 1.5B and 7B DeepSeek-R1-Distill-Qwen backbones. These are relatively small models compared to modern reasoning LLMs (e.g., 30B–70B). It remains unclear whether the proposed framework maintains its benefits or efficiency when scaled up. Larger-scale validation would strengthen the generality claim.
>
>
> > Q3 How stable is training when applied to larger models (e.g., ≥30B parameters)?
>
> Thank you for raising this important point.
>
>
> In our experiment, we utilized model scales (1.5B and 7B) consistent with recent works in this filed (e.g., AdaptThink, DAST and ShorterBetter).
>
>
> Currently, due to computational constraints, we are unable to test on backbones exceeding 30B parameters. However, in future work, once adequate computational resources become available, we plan to expand the application of RazorReward-RL to larger-scale backbones (e.g., DeepSeek-R1-Distill-32B/70B and similar models).
>
> ---
>
> > Weekness  (5) The benchmarks focus almost exclusively on mathematical and scientific reasoning. It is uncertain whether the same reward dynamics hold for open-domain, commonsense, or multimodal reasoning, where the notion of “minimal sufficient reasoning” is less clearly defined.
>
>
> > Q2  Can the approach be extended to open-ended or dialogue reasoning tasks, where “minimal sufficient” reasoning is ill-defined?
>
> Thank you for this constructive suggestions.
>
>
> In response, we conducted supplemental experiments on HotpotQA, a challenging open-domain, multi-document question-answering dataset. This task requires models to retrieve, synthesize, and reason across multiple documents.
>
> **Table 4. Reasoning performance on the HotpotQA dataset (EM and F1 scores).**
> | Model              | EM   | F1   |
> |--------------------|----------|----------|
> | RazorReward-RL     | 0.1688   | 0.2313   |
> | ShorterBetter      | 0.0859   | 0.1381   |
> | AdaptThink         | 0.0621   | 0.0978   |
> | ShorterBetter      | 0.0483   | 0.0752   |
> | Vanilla            | 0.0309   | 0.0530   |
>
> The results demonstrated that RazorReward-RL achieved an Exact Match (EM) score of 16.88% and an F1 score of 23.13% on HotpotQA, outperforming all baseline methods by over 10 percentage points on average. This proves our approach’s effectiveness in the cross-document, open-domain QA task beyond mathematical or scientific tasks.
>
>
> We acknowledge the challenge of defining “minimal sufficient reasoning”  in tasks without definitive ground-truth answers (e.g., open-ended dialogue). To extend our method to such cases,  frontier LLMs (e.g., Gemini-3) can be used to generate high-quality reference answers. Then, semantic similarity can measure a response's correctness. This remains an avenue for future work.

---

### Official Review · Reviewer_sB1h · 2025-10-28

**Soundness:** 3
**Presentation:** 3
**Contribution:** 3
**Rating:** 6
**Confidence:** 2

**Summary:**

The paper proposes RazorReward-RL, a reinforcement learning framework designed to reduce redundant reasoning (overthinking) in large language models’ chain-of-thought (CoT). It introduces RazorReward, a novel reward function that enforces sharp penalties for both under- and over-reasoning relative to the minimal sufficient reasoning length. The method segments CoT into semantically coherent reasoning blocks using a lexicon-based approach and applies a combined reward of correctness, format, and length efficiency. Experiments on six reasoning benchmarks (GSM8K, MATH500, AIME24/25, AMC23, GPQA-D) using DeepSeek-R1-Distill-Qwen-7B/1.5B show accuracy gains of 8–9% and token reductions of 38–44%, achieving the best accuracy-efficiency trade-off compared to SOTA baselines such as DAST and ShorterBetter. Ablation and out-of-distribution tests further confirm that RazorReward-RL yields more concise, focused, and generalizable reasoning without sacrificing correctness, offering a principled solution to the LLM overthinking problem.

**Strengths:**

* Novel Reward Design (RazorReward): The proposed length-sensitive, sign-sharpened reward sharply separates optimal from suboptimal reasoning. It improves upon prior smooth reward schemes (e.g., DAST, ShorterBetter) by introducing exponential decay and fixed post-optimum penalties that tightly constrain CoT length.

* Balanced Trade-off Between Accuracy and Efficiency: Results show RazorReward-RL reduces average token count by ~40% while achieving up to +9% absolute accuracy gain — a compelling demonstration that reasoning brevity need not compromise correctness.

* OOD Generalization Evidence: The additional experiments on MMLU and GPQA-Diamond show transferability beyond math domains, supporting claims of generalizability.

**Weaknesses:**

* Reward Sensitivity & Hyperparameter Study Missing: No ablation or sensitivity analysis is provided on the exponential decay constants or penalty weights (especially within Eq. (2)). This omission limits understanding of stability and optimality during RL training.

* Limited Backbone Diversity: Experiments rely solely on Qwen-Distill backbones (7B and 1.5B). The framework’s behavior on instruction-tuned, general-purpose, or reasoning-specialized models (e.g., DeepSeek-R1-Distill-70B, OpenAI-o1) remains untested.

**Questions:**

See the Weaknesses

---

> ### Author Response · Authors · 2025-11-28
> **Response to Reviewer sB1h**
>
> > 1. Reward Sensitivity & Hyperparameter Study Missing: No ablation or sensitivity analysis is provided on the exponential decay constants or penalty weights (especially within Eq. (2)). This omission limits understanding of stability and optimality during RL training.
>
> Thank you for raising this critical point regarding the need for sensitivity analysis, specifically the hyperparameter in Eq. (2) (i.e., $\alpha$).
>
>
> To address this, we performed a sensitivity analysis on $\alpha$. As requested by Reviewer SCZ5, this analysis was performed under our revised experimental settings (training length: 4k tokens, inference length: 16k tokens). The key results are summarized in the table below:
>
>
>
> **Table 1: Sensitivity analysis results for the decay coefficient α.**
>
>
> | Model     | Avg Acc↑ | Avg Tok↓ | Acc Delta | Tok Delta | AES  | aime_2024 Acc↑ | aime_2024 Tok↓ | aime_2025 Acc↑ | aime_2025 Tok↓ | amc2023 Acc↑ | amc2023 Tok↓ | gqpa_diamond Acc↑ | gqpa_diamond Tok↓ | gsm8k Acc↑ | gsm8k Tok↓ | math500 Acc↑ | math500 Tok↓ |
> |-----------|----------|----------|-----------|-----------|------|----------------|----------------|----------------|----------------|--------------|--------------|-------------------|-------------------|------------|------------|--------------|--------------|
> | α = 1/2   | 0.635    | 4,017    | 5.54%     | 25.45%    | 0.54 | 0.433          | 6,450          | 0.300          | 6,839          | 0.875        | 3,591        | 0.399             | 4,436             | 0.929      | 708        | 0.872        | 2,076        |
> | α = 2/3   | 0.613    | 4,532    | 3.42%     | 15.90%    | 0.34 | 0.367          | 7,542          | 0.267          | 7,809          | 0.900        | 3,364        | 0.359             | 5,409             | 0.923      | 814        | 0.866        | 2,252        |
>
> Setting $\alpha=1/2$  yielded an average accuracy (avg_acc) of 0.6347 and an average reasoning length (avg_length) of 4016.8. Increasing $\alpha$ to 2/3 resulted in a slightly lower average accuracy (0.6134) but a significantly longer average reasoning length (4531.7). This demonstrates the substantial influence $\alpha$ has on the trade-off between model accuracy and reasoning efficiency. Thus, the $\alpha$ should be carefully selected to optimize performance. We used $\alpha=1/2$ following the S-GRPO paper[1].
>
> **Reference**:
>
> [1] Muzhi Dai, Chenxu Yang, and Qingyi Si. S-GRPO: early exit via reinforcement learning in reasoningmodels. CoRR, abs/2505.07686, 2025b. doi: 10.48550/ARXIV.2505.07686. URL https://doi.org/10.48550/arXiv.2505.07686.
>
>
> > 2. Limited Backbone Diversity: Experiments rely solely on Qwen-Distill backbones (7B and 1.5B). The framework’s behavior on instruction-tuned, general-purpose, or reasoning-specialized models (e.g., DeepSeek-R1-Distill-70B, OpenAI-o1) remains untested.
>
> Thank you for this valuable suggestion.
>
>
> We primarily conducted experiments using two open-source backbone models: DeepSeek-R1-Distill-7B and 1.5B. This choice is to align with recent works on mitigating overthinking in LLMs, like AdaptThink and ShorterBetter.
>
> We further validated our approach using Qwen2.5-1.5B-Instruct for your concern to the instruction-tuned model. The results are presented in the table below:
>
> **Table 2. Performance of RazorReward-RL on Qwen2.5-1.5B-Instruct model across various datasets (accuracy and token usage).**
>
> | Model            | aime_2024 Acc↑ | aime_2024 Tok↓ | aime_2025 Acc↑ | aime_2025 Tok↓ | amc2023 Acc↑ | amc2023 Tok↓ | gqpa_diamond Acc↑ | gqpa_diamond Tok↓ | gsm8k Acc↑ | gsm8k Tok↓ | math500 Acc↑ | math500 Tok↓ | avg_acc | avg_tok | Acc_delta | Tok_delta | AES   |
> |------------------|----------------|----------------|----------------|----------------|--------------|--------------|-------------------|-------------------|------------|------------|--------------|--------------|---------|---------|-----------|-----------|-------|
> | Vanilla   | 0.033          | 1499           | 0.000          | 1299           | 0.225        | 839          | 0.152             | 762               | 0.613      | 279        | 0.462        | 662          | 0.248   | 768     | 0.00%     | 0.00%     | 0.000 |
> | RazorReward-RL | 0.033      | 933            | 0.033          | 1010           | 0.200        | 754          | 0.288             | 654               | 0.739      | 265        | 0.442        | 497          | 0.289   | 685     | 4.18%     | 10.78%    | 0.614 |
>
> Our method remained effective, improving Vanilla accuracy by 4.18%, reducing token consumption by 10.78%, and achieving an AES score of 0.614.
>
> We appreciate your perspective on evaluating with larger backbone like DeepSeek-R1-Distill-70B and OpenAI-o1. However, testing our method on these models is currently beyond our resource capacity. We acknowledge this gap and will extend our evaluation to broader model size in future work.

---

### Official Review · Reviewer_SCZ5 · 2025-10-30

**Soundness:** 2
**Presentation:** 3
**Contribution:** 2
**Rating:** 2
**Confidence:** 4

**Summary:**

This paper proposes **RazorReward-RL**, a reinforcement learning framework designed to minimize redundant reasoning in large language models. The key idea is **RazorReward**, a sharp reward mechanism that penalizes both over-reasoning relative to the minimal sufficient chain of thought (CoT). The method segments reasoning into semantically meaningful blocks and applies fine-grained rewards to encourage concise yet complete reasoning. Experiments on six reasoning benchmarks show that RazorReward-RL improves accuracy by around 8–9% while reducing token usage by over 38%, achieving a strong balance between accuracy and efficiency.

**Strengths:**

Strengths:
1. Experiments across multiple benchmarks demonstrate consistent improvements in both accuracy and efficiency.
2. The proposed approach is simple to implement and could generalize well to other RL-based reasoning frameworks.

**Weaknesses:**

Weaknesses:
1. The evaluation setup limits models to a 2048-token context window, which is not a fair comparison, as most recent reasoning models are trained or evaluated with much longer contexts. Efficient reasoning should emphasize the trade-off between accuracy and efficiency rather than constraining prior methods with artificially short contexts.
2. The paper encourages shorter reasoning traces even on hard questions that the model fails to solve. The motivation for this design is unclear—if a model cannot answer a hard problem correctly with extended reasoning, it seems counterintuitive that reducing reasoning length would help.
3. The comparison mainly focuses on quantitative improvements but provides limited **qualitative analysis** of reasoning behaviors (e.g., what kinds of redundant reasoning are trimmed).

Statement:
I assign a score of 2 at this stage mainly due to concerns about the current evaluation setup, which I believe has major issues affecting the fairness and validity of the results. However, if the authors can provide a more reasonable and comprehensive evaluation—demonstrating fair comparisons and stronger evidence for the claimed efficiency gains—I would be willing to reconsider and raise my score, potentially to a positive rating. I will revisit this paper carefully once the evaluation concerns are properly addressed.

**Questions:**

See weakness

---

> ### Author Response · Authors · 2025-11-28
> **Response to Reviewer SCZ5 (Part 1)**
>
> > 1. The evaluation setup limits models to a 2048-token context window, which is not a fair comparison, as most recent reasoning models are trained or evaluated with much longer contexts. Efficient reasoning should emphasize the trade-off between accuracy and efficiency rather than constraining prior methods with artificially short contexts.
>
> Thank you for this insightful comment.
>
>
> **We chose the 2048-token context specifically to meet the constraints in Baidu's search-based AI Q&A system, where latency significantly impacts user experience. Baidu's historical data also indicates that the majority of responses in this scenario fall below 2048 tokens.**
>
>
> However, we appreciate your suggestion on evaluating with longer contexts. **To address your concern, we conducted evaluations using a 16k-token context.** This setting matches the inference settings of recent works including Deer, Shorterbetter, S-GRPO, AdaptiveLLM, Thinkless and AdaptThink.
>
>
> Results are shown in the following table:
>
>
> **Table 4. Performance with extended reasoning length (16k tokens) on six benchmark datasets.**
>
>
> | Model            | aime_2024 Acc↑ | aime_2024 Tok↓ | aime_2025 Acc↑ | aime_2025 Tok↓ | amc2023 Acc↑ | amc2023 Tok↓ | gqpa_diamond Acc↑ | gqpa_diamond Tok↓ | gsm8k Acc↑ | gsm8k Tok↓ | math500 Acc↑ | math500 Tok↓ | Avg_Acc | Avg_Tok | Acc_delta | Tok_delta | AES   |
> |------------------|----------------|----------------|----------------|----------------|--------------|--------------|-------------------|-------------------|------------|------------|--------------|--------------|---------|---------|-----------|-----------|-------|
> | RazorReward-RL   | 0.433          | 6450           | 0.300          | 6839           | 0.875        | 3591         | 0.399             | 4436              | 0.929      | 708        | 0.872        | 2076         | 0.635   | 4017    | 5.54%     | 25.45%    | 0.606 |
> | L1-Max           | 0.200          | 2125           | 0.200          | 1971           | 0.675        | 1448         | 0.338             | 1287              | 0.582      | 527        | 0.722        | 1060         | 0.453   | 1403    | -12.65%   | 73.96%    | -0.352|
> | Vanilla          | 0.400          | 10758          | 0.433          | 9340           | 0.825        | 4869         | 0.237             | 4568              | 0.784      | 423        | 0.796        | 2371         | 0.579   | 5388    | -         | -         | 0.000 |
> | DAST             | 0.567          | 9146           | 0.400          | 10416          | 0.800        | 5283         | 0.197             | 7644              | 0.913      | 837        | 0.866        | 2674         | 0.624   | 6000    | 4.45%     | -11.35%   | -0.063|
> | ShorterBetter    | 0.300          | 4175           | 0.200          | 3605           | 0.650        | 1252         | 0.318             | 1251              | 0.764      | 181        | 0.688        | 749          | 0.487   | 1869    | -9.25%    | 65.32%    | -0.088|
> | AdaptThink       | 0.533          | 8547           | 0.333          | 7683           | 0.875        | 4144         | 0.313             | 4696              | 0.821      | 392        | 0.868        | 1867         | 0.624   | 4555    | 4.47%     | 15.46%    | 0.401 |
>
> The results demonstrate RazorReward-RL's efficiency in this setting. Specifically, it achieves **5.5% higher average accuracy than Vanilla while surpassing the SOTA DAST by 1.08%. Moreover, it reduces average token consumption by 25% compared to Vanilla**. Crucially, RazorReward-RL achieves the **optimal AES score** among all compared methods.
>
>
> We will include these results into the revised manuscript.

---

> ### Author Response · Authors · 2025-11-28
> **Response to Reviewer SCZ5 (Part 2)**
>
> > 2. The paper encourages shorter reasoning traces even on hard questions that the model fails to solve. The motivation for this design is unclear—if a model cannot answer a hard problem correctly with extended reasoning, it seems counterintuitive that reducing reasoning length would help.
>
> Thank you for this insightful feedback.
>
>
> We penalize long reasoning for hard questions based on the observation: **when confronting problems beyond the model's capabilities, longer reasoning sequences yield marginal accuracy gains. We aim to reduce model's token waste on these questions, while not meaning this can help answer correctly**.
>
>
> To demonstrate this, we compared two reasoning methods across 101 hard questions where the model consistently failed across the truncated CoTs. These questions are sampled from AIME24 (8), AIME25 (19) and MATH500 (74), using the DeepSeek-R1-Distill-Qwen-7B model.
>
>
> The first method employed concise reasoning ("Please reason concisely"), while the second appended initial reasoning traces to new prompts with "Please continue reasoning," forcing iterative extensions. The results are presented in the table below:
>
> **Table 2. Accuracy and token usage for concise and extended reasoning on hard questions.**
> | Mode         | Dataset | Tok        | Acc   | Dataset     | Length    | Acc    | Avg Length | Avg Acc |
> |--------------|---------|------------|---------|-------------|-----------|----------|------------|-----------|
> | Concise      | aime    | 4018.52    | 0.00%   | math_hard   | 1,362.58  | 72.97%   | 2,072.58   | 0.53      |
> | 1st Long     | aime    | 8173.41    | 0.00%   | math_hard   | 2,234.18  | 72.97%   | 3,821.89   | 0.53      |
> | 2nd Long     | aime    | 13123.30   | 3.70%   | math_hard   | 3,905.17  | 72.97%   | 6,369.42   | 0.54      |
>
> It can be found that increasing reasoning length fails to yield effective accuracy gains while incurring disproportionate computational costs:
> - When expanding reasoning length from 4,096 tokens (Concise Reasoning) to 8192 tokens (first extended reasoning cycle), **no significant difference in accuracy emerged**.
> - Upon introducing a second reasoning cycle, accuracy improved by merely ~1% (i.e., 1/101 correct answer gained). This consumed 2548 additional tokens (+66% over the first cycle).
> These results motivate our design of optimizing the model toward shorter reasoning when the questions are hard or unsolvable.
>
> > 3. The comparison mainly focuses on quantitative improvements but provides limited qualitative analysis of reasoning behaviors (e.g., what kinds of redundant reasoning are trimmed).
>
> Thank you for this insightful comment.
>
>
> Regarding the qualitative analysis of reasoning behaviors, we presented a “Reasoning Quality Analysis” in Section 4.3.1 of our manuscript. Specifically, our quantitative analysis reveals that the RazorReward-RL increases the proportion of **core reasoning tokens (64.5% to 76.6%) while reducing tokens on verification and exploratory thinking (31.7% to 18.4%).**
>
>
> Through manual inspection, we identified six major categories of redundant reasoning behaviors that were substantially reduced after training, including *repeated verification,over-exploration,over-detailing,self-doubt loops, hypothetical exploration,ritualistic steps.*
>
>
> Our analysis reveals that RazorReward-RL exhibits consistent improvement trends across tasks of varying difficulty (GSM8K and AMC23). In particular, the model’s reasoning trace evolves from lengthy, repetitive exploratory thinking into a more efficient and well-structured reasoning chain.The improvements are reflected in the following aspects:
>
>
> - **More concise expression**: Redundant repetition, trial‑and‑error reasoning, and meta‑linguistic commentary are significantly reduced,  allowing the model to focus more on core computational steps.
>
>
> - **Enhanced determinism**: The model shows fewer invalid exploratory attempts and less uncertain expression, producing clearer and more coherent reasoning chains.
>
>
> The focus of improvement varies with task difficulty:
>
>
> - GSM8K Dataset(Easy) :  **RL training primarily helps the model avoid redundant calculations and trivial mistakes.**
>
>
> - AMC23 (Hard):  **The model demonstrates stronger capability in handling structurally complex problems by decomposing them into logically rigorous mathematical steps and reducing erroneous explorations.**
>
>
> These findings indicate that RazorReward-RL preserves essential reasoning while pruning inefficient or redundant reasoning with high efficacy.

---

### Official Review · Reviewer_Xhc2 · 2025-10-31

**Soundness:** 2
**Presentation:** 3
**Contribution:** 2
**Rating:** 4
**Confidence:** 4

**Summary:**

This paper aims to solve the "overthinking" problem for large reasoning models. The authors propose the RazorReward-RL framework. This framework segments the CoT into semantically meaningful blocks to construct training samples and mitigates overthinking through a novel reward mechanism (RazorReward). For hard queries, it penalizes unnecessary steps and encourages abstention; for medium-difficulty queries, it rewards only the paths that exactly match the "minimal sufficient CoT" and heavily penalizes both under- and over-reasoning. Experimental results show that RazorReward-RL outperforms previous methods on multiple math reasoning benchmarks, improving model accuracy while significantly reducing average token usage.

**Strengths:**

1. The authors propose a novel, annotation-free difficulty classification and reward mechanism. The paper devises a novel CoT truncation sampling framework and, by analyzing the answer correctness at different truncation lengths, automatically classifies queries into different difficulty levels. This classification is adaptive and does not require additional human annotation. Based on this classification, RazorReward can design highly targeted reward functions.


2. Experimental results demonstrate that it can achieve good results in accuracy with fewer tokens.

**Weaknesses:**

1. The reward mechanism might encourage guessing rather than sufficient reasoning. The design of RazorReward is to heavily reward the first truncated CoT that can guess the correct answer while severely penalizing all subsequent steps, assuming the remaining to be redundant. This is not very promising from some perspectives. First, the shortest CoT that can guess the correct answer does not seem to necessarily align perfectly with a logically complete "minimal sufficient reasoning." Second, many studies have shown that "Verification" steps are crucial for improving the final accuracy of complex reasoning. Although excessive verification is redundant, the paper's reward function penalizes all steps for verification, which discourages the model from performing any form of verification, even if it is helpful for making the reasoning robust.

2. In terms of experimental setup, the current experiments for training and testing are limited to mathematical reasoning, and the models used are distilled versions of DeepSeek-R1. More experiments and models are needed to prove whether the lexicon-based structural reasoning blocks possess generalization capabilities, rather than being limited to specific tasks or the writing style of DeepSeek-R1.

**Questions:**

1. In the methodology section (3.1.1), the paper emphasizes that one of its core contributions is using lexicon-based "semantic chunking" to construct samples. However, in the subsequent "Reasoning Quality Analysis" (4.3.1) , the evaluation suddenly switches to the "line-level". Why not continue to use the previously defined "semantic blocks" as the basic unit of analysis?

2. Regarding the "encouragement of abstention", is the model allowed to generate a specific "I don't know" or "unsolvable" token to receive a reward?

3. What is the "Vanilla" setting in the table?

4. Does the reward designed by the authors help improve the model's accuracy? Why does it achieve a significant accuracy improvement compared to Basic RL?

---

> ### Author Response · Authors · 2025-11-28
> **Response to Reviewer Xhc2 (Part 1)**
>
> Thank you for your valuable feedback ! Our response is as follows:
> > 1. The reward mechanism might encourage guessing rather than sufficient reasoning. The design of RazorReward is to heavily reward the first truncated CoT that can guess the correct answer while severely penalizing all subsequent steps, assuming the remaining to be redundant. This is not very promising from some perspectives. First, the shortest CoT that can guess the correct answer does not seem to necessarily align perfectly with a logically complete "minimal sufficient reasoning."
>
> Thank you for raising this critical point regarding the potential for guessing versus sufficient reasoning.
>
> To address your concern, we assessed reasoning sufficiency across the Math500 dataset using Qwen3-Max as a judge. Our analysis focused on four key dimensions: step validity, coherence, conclusion support and logical completeness.
>
> - In terms of training samples, we performed logical analysis on the first correct sample (i.e., the first truncated CoT that can guess the correct answer) obtained during rollout sampling. The results show that the logical compelteness of these samples shows an average score of 0.95 out of a maximum of 1.00. Each individual metric—including step validity, coherence, and conclusion support—scored around 4.85 out of a maximum of 5. This demonstrates that the samples collected during the training process possess complete and sufficient logical reasoning, rather than being derived from mere guessing.
>
> - With regard to the model’s inference results, we evaluated logical completeness across different difficulty levels.  The results are presented in the following table:
>
> **Table 1. Reasoning Quality Evaluation on Math500 (Hard Problems, 100 samples).**
>
> | Model            | Step Validity | Logical Coherence | Conclusion Support | Overall Logical Completeness  | Token |
> |------------------|--------------|-------------------|--------------------|--------------------------|-------|
> | Vanilla          | 4.72         | 4.74              | 4.60               | 0.885                    | 1435|
> | RazorReward RL   | 4.66         | 4.68              | 4.59               | 0.885                    | 1066|
> | DAST             | 4.86         | 4.86              | 4.78               | 0.935                    | 1707|
> | ShorterBetter    | 4.41         | 4.49              | 4.35               | 0.805                    | 446 |
>
> ---
>
> **Table 2. Reasoning Quality Evaluation on Math500 (Medium Difficulty, 100 samples).**
>
> | Model            | Step Validity | Logical Coherence | Conclusion Support | Overall Logical Completeness  | Token |
> |------------------|--------------|-------------------|--------------------|--------------------------|-------|
> | Vanilla          | 4.94         | 4.94              | 4.96               | 0.985                    | 1129|
> | RazorReward RL   | 4.92         | 4.92              | 4.92               | 0.970                    | 884 |
> | DAST             | 4.98         | 4.98              | 4.98               | 0.995                    | 1413|
> | ShorterBetter    | 4.65         | 4.70              | 4.65               | 0.905                    | 348 |
>
> ---
>
> **Table 3. Reasoning Quality Evaluation on Math500 (Easy Problems, 50 samples).**
>
> | Model            | Step Validity | Logical Coherence | Conclusion Support | Overall Logical Completeness  | Token |
> |------------------|--------------|-------------------|--------------------|--------------------------|-------|
> | Vanilla          | 5.00         | 5.00              | 5.00               | 1.000                    | 816 |
> | RazorReward RL   | 4.88         | 4.88              | 4.88               | 0.970                    | 668 |
> | DAST             | 5.00         | 5.00              | 5.00               | 1.000                    | 1102|
> | ShorterBetter    | 4.80         | 4.80              | 4.82               | 0.910                    | 154 |
>
> ---
> - For Mid and Easy problems, RazorReward-RL achieved about 97% logical completeness, closely matching baselines like DAST and Vanilla. In Hard problems, models stopped earlier by design, producing shorter reasoning than baselines like DAST and Vanilla, while still preserving over 89% logical completeness.
> These findings confirm that our approach effectively eliminates redundant steps without compromising the core logical progression.

---

> ### Author Response · Authors · 2025-11-28
> **Response to Reviewer Xhc2 (Part 2)**
>
> > 2. Second, many studies have shown that "Verification" steps are crucial for improving the final accuracy of complex reasoning. Although excessive verification is redundant, the paper's reward function penalizes all steps for verification, which discourages the model from performing any form of verification, even if it is helpful for making the reasoning robust.
>
> Thank you for your insightful comment.
>
> **We wish to clarify that RazorReward-RL’s reward mechanism does not indiscriminately penalize all verification steps. It specifically suppresses redundant verification behaviors that do not contribute to correctness – that is, repeated checking when the model fails to arrive at the correct answer.** This is guaranteed by the correctness component in our reward function (Line 205-220).
>
> To demonstrate this, we analyzed verification behavior on the Math500 and AMC2023 datasets using Qwen3-max as a judge (see table below).
>
> **Table 4: Performance and Verification Step Analysis of Different Methods on the MATH500 and AMC2023 Datasets**
>
> | Model                      | Dataset  | Rate_verification | Acc   | Token |
> |----------------------------|----------|------------------|-------|-------|
> | RazorReward-RL             | math500  | 0.39             | 81.2  | 1088  |
> | ShorterBetter              | math500  | 0.15             | 76.6  | 492   |
> | DAST                       | math500  | 0.94             | 80.5  | 1802  |
> | AdaptThink                 | math500  | 0.43             | 81.7  | 1296  |
> | Vanilla                    | math500  | 0.43             | 73.5  | 1580  |
> |                            |          |                  |       |       |
> | RazorReward-RL             | amc2023  | 0.325            | 74.7  | 1656  |
> | ShorterBetter              | amc2023  | 0.3              | 68.6  | 1000  |
> | DAST                       | amc2023  | 0.75             | 66.1  | 2709  |
> | AdaptThink-7B-delta0.05    | amc2023  | 0.6              | 72.8  | 2240  |
> | Vanilla                    | amc2023  | 0.6              | 59.5  | 2739  |
>
> After RazorReward-RL training, the overall frequency of verification steps significantly decreased. Specifically, the proportion of samples containing verification steps decreased **from 43% to 39%** on the Math500 dataset and **from 60% to 32.5%** on the AMC2023 dataset. Nevertheless, the overall accuracy remained higher than that of prior baseline models.
>
>
> A more detailed analysis on the Math500 dataset reveals that, after training, the proportion of correct samples involving verification steps increased **from 29% to 37%**, while that of incorrect samples decreased **from 54% to 43%**.
>
> These findings demonstrate that RazorReward-RL effectively distinguishes between beneficial and redundant verification behaviors, reducing unnecessary checks while retaining essential validation.
>
>
> **Table 5: Further Analysis of Verification Steps in Relation to Answer Correctness on the MATH500 Dataset**
> | Model          | correct_type | rate_verification |
> |----------------|--------------|-------------------|
> | RazorReward-RL | correct      | 37.14%            |
> | Vanilla        | correct      | 29.79%            |
> | RazorReward-RL | incorrect    | 43.33%            |
> | Vanilla        | incorrect    | 54.72%            |

---

> ### Author Response · Authors · 2025-11-28
> **Response to Reviewer Xhc2 (Part 3)**
>
> > 3. In terms of experimental setup, the current experiments for training and testing are limited to mathematical reasoning, and the models used are distilled versions of DeepSeek-R1. More experiments and models are needed to prove whether the lexicon-based structural reasoning blocks possess generalization capabilities, rather than being limited to specific tasks or the writing style of DeepSeek-R1.
>
> Thank you for your valuable suggestion.
>
>
> We wish to clarify that in **Section 4.3.1** of our manuscript, the tested MMLU and GPQA benchmarks are QA tasks not limited to mathematical reasoning.
>
>
> To further demonstrate the generalizability of our method, we conduct additional testing on **HotpotQA**, a benchmark commonly used in search domains. Our evaluation included top-performing baseline models from our main experiments (ShorterBetter, DAST, AdaptThink) alongside the Vanilla model. The results are presented in the table below:
>
> **Table 6. Reasoning performance on the HotpotQA dataset (EM and F1 scores).**
> | Model              | EM   | F1   |
> |--------------------|----------|----------|
> | RazorReward-RL     | 0.1688   | 0.2313   |
> | ShorterBetter      | 0.0859   | 0.1381   |
> | AdaptThink         | 0.0621   | 0.0978   |
> | ShorterBetter      | 0.0483   | 0.0752   |
> | Vanilla            | 0.0309   | 0.0530   |
>
> On the HotpotQA dataset, RazorReward-RL **achieves 16.88% EM and 23.13% F1 score**, significantly outperforming baseline models like ShorterBetter (8.59%), AdaptThink (6.21%), and DAST (4.83%). These results indicate that RazorReward-RL maintains effectiveness in this domain.
>
>
> **We further validated our approach using Qwen2.5-1.5B-Instruct, a model from a different family than DeepSeek**. The results are presented in the table below:
>
> **Table 7. Performance of RazorReward-RL on Qwen2.5-1.5B-Instruct model across various datasets (accuracy and token usage).**
> | Model            | aime_2024 Acc↑ | aime_2024 Tok↓ | aime_2025 Acc↑ | aime_2025 Tok↓ | amc2023 Acc↑ | amc2023 Tok↓ | gqpa_diamond Acc↑ | gqpa_diamond Tok↓ | gsm8k Acc↑ | gsm8k Tok↓ | math500 Acc↑ | math500 Tok↓ | avg_acc | avg_tok | Acc_delta | Tok_delta | AES   |
> |------------------|----------------|----------------|----------------|----------------|--------------|--------------|-------------------|-------------------|------------|------------|--------------|--------------|---------|---------|-----------|-----------|-------|
> | Vanilla   | 0.033          | 1499           | 0.000          | 1299           | 0.225        | 839          | 0.152             | 762               | 0.613      | 279        | 0.462        | 662          | 0.248   | 768     | 0.00%     | 0.00%     | 0.000 |
> | RazorReward-RL | 0.033      | 933            | 0.033          | 1010           | 0.200        | 754          | 0.288             | 654               | 0.739      | 265        | 0.442        | 497          | 0.289   | 685     | 4.18%     | 10.78%    | 0.614 |
>
>
> Our method remained effective, improving Vanilla accuracy by 4.18%, reducing token consumption by 10.78%, and achieving an AES score of 0.614.
>
> > 4. Regarding the "encouragement of abstention", is the model allowed to generate a specific "I don't know" or "unsolvable" token to receive a reward?
>
> Regarding your question about abstention encouragement:
> The model is trained to provide concise answers rather than avoid responding. We do not reward the generation of specific abstention tokens (e.g., "I don’t know" or "unanswerable").
> Empirically, we observed no cases where the model produced such tokens, even for hard questions.  Instead, it consistently attempts to provide an answer.
>
> > 5. What is the "Vanilla" setting in the table?
>
> "Vanilla" refers to the initial DeepSeek-R1-Distill-Qwen model, with no further fine-tuning applied.

---

> ### Author Response · Authors · 2025-11-28
> **Response to Reviewer Xhc2 (Part 4)**
>
> > 6. In the methodology section (3.1.1), the paper emphasizes that one of its core contributions is using lexicon-based "semantic chunking" to construct samples. However, in the subsequent "Reasoning Quality Analysis" (4.3.1) , the evaluation suddenly switches to the "line-level". Why not continue to use the previously defined "semantic blocks" as the basic unit of analysis?
>
>
> Thank you for this insightful question regarding our shift in analysis units.
>
>
> **We employ line-level granularity in Reasoning Quality Analysis because it aligns more closely with LLMs' native reasoning evaluation capabilities, thus enhancing objectivity.** Lexicon-based semantic chunking, although efficient for constructing training samples, does not offer this advantage compared to line-level granularity.
>
>
> To address your concern, we evaluated reasoning quality using semantic blocks. The results are shown below:
>
> **Table 8. Semantic block-level reasoning behavior analysis results.**
>
> | Model           | purity | logical_quality | coherence | Core Reasoning (%) | Verification & Exploration (%) | Pivotal Reasoning | Productive Elaboration & Calculation | Verification & Self-Correction | Exploring Alternatives | Non-Substantive Statement |
> |-----------------|------------|---------------------|---------------|--------------------|-------------------------------|-------------------|--------------------------------------|-------------------------------|-----------------------|---------------------------|
> | RazorReward-RL  | 4.5377     | 4.7138              | 4.7516        | 85.76%             | 11.14%                        | 25.37%            | 60.39%                               | 5.87%                         | 5.27%                 | 3.10%                     |
> | ShorterBetter   | 4.3770     | 4.4409              | 4.5272        | 82.59%             | 14.33%                        | 27.12%            | 55.47%                               | 6.59%                         | 7.74%                 | 3.08%                     |
> | Vanilla         | 4.4715     | 4.7595              | 4.7247        | 74.65%             | 23.71%                        | 18.67%            | 55.98%                               | 13.05%                        | 10.66%                | 1.64%                     |
> | DAST            | 4.4272     | 4.7120              | 4.6764        | 69.67%             | 27.77%                        | 19.58%            | 50.09%                               | 12.46%                        | 15.31%                | 2.57%                     |
>
> We found that our method achieved **high coverage (85.76%) in Pivotal Reasoning and Productive Elaboration & Calculation, significantly outperforming all baselines (by 3.2, 16, and 11.1 percentage points, respectively). Conversely, Verification & Self-Correction and Exploring Alternatives occurred at a lower rate under our method compared to baselines (27%, 23%, 14%).
> Crucially, semantic-block analysis reinforces the core findings from our line-level evaluation**. Both demonstrate the superior capacity of our method in preserving core reasoning steps while reducing redundant ones.
>
> > 7. Does the reward designed by the authors help improve the model's accuracy? Why does it achieve a significant accuracy improvement compared to Basic RL?
>
> Yes, our reward function improve model accuracy through a correctness component (Line 205-220). The component guides the model towards correct CoTs during training by comparing and favoring them over incorrect ones. This helps boost model accuracy.
>
>
> The improvement over Basic RL stems primarily from two key advantages of our semantic truncation method compared to its random truncation:
>
>
> 1. **Constructing training samples via semantic truncation yields a higher proportion of positive samples compared to random truncation**. Consequently, the model receives more positive reward signals during RL training. This was empirically validated: inference on the Math500 dataset using CoTs generated via semantic truncation showed a 2.75% higher accuracy than those using random truncation.
>
>
> 2. **The CoT paths generated using semantic truncation exhibit higher reasoning quality than those using random truncation**. Specifically, using the qwen3-max model as judge, we find that semantically truncated CoTs score higher on logical completeness and conclusion support.
>
>
> These effects lead to improvements over Basic RL.

---

### Note · Authors · 2025-12-04

**Comment:**

We sincerely thank all reviewers, and area chairs for their valuable feedback and active engagement throughout the review process. **Our team thoroughly considered each reviewer’s comments, and during the rebuttal phase, we provided detailed responses to every point raised, including new experiments, ablation studies, and additional qualitative analyses as requested.**


We hope that our responses have helped clarify any misunderstandings and addressed methodological concerns, while the added HotpotQA experiments, expanded backbone models, qualitative results, and sensitivity analyses further demonstrate the robustness and generality of our approach.

At the same time, we are also deeply grateful for the valuable suggestions provided by the reviewers to help improve the quality of our manuscript, which are crucial for the further refinement of this line of research.


However, after careful consideration, we have decided to withdraw our paper at this stage.


In summary, we sincerely appreciate the constructive feedback provided by the reviewers. The issues and insights raised will play a crucial role in shaping and strengthening the next iteration of this work.

**Withdrawal Confirmation:**

I have read and agree with the venue's withdrawal policy on behalf of myself and my co-authors.